# OpenReview forum: "Soft Prompts Go Hard: Steering Visual Language Models with Hidden Meta-Instructions"
_ICLR.cc/2025/Conference — Submitted to ICLR 2025_

### Official Review · Reviewer_YgaV · 2024-10-31

**Soundness:** 2
**Presentation:** 3
**Contribution:** 2
**Rating:** 5
**Confidence:** 4

**Summary:**

This paper proposes a new attack objective in which the output text remains consistent with the input images but adopts an adversary-chosen style, sentiment, or point of view. The adversarial optimization is applied to the input image, ensuring that the modifications are imperceptible to humans. Experiments demonstrate that images containing hidden meta-instructions achieve significantly higher success rates compared to those with explicit instructions. This attack highlights a practical risk, as it enables the dissemination of seemingly coherent but misleading information.

**Strengths:**

1. The focus on the dissemination of seemingly coherent misinformation is highly practical and addresses a significant real-world concern.

2. The evaluation is thorough, including robustness testing against JPEG compression as a defense (which I suggest moving to the main text, given its practicality in everyday use) and examining the transferability of the attack across different vision-language models (VLMs).

**Weaknesses:**

1. A NeurIPS 2024 paper [1] also explores the dissemination of seemingly coherent misinformation in visual language models, but through the lens of data poisoning. While this paper focuses on test-time adversarial attacks, it would be beneficial to discuss the key differences between test-time attacks and training-time poisoning, and in what scenarios each is more practical, given the similarity in objectives between the two papers.

2. The evaluation of image semantics preservation seems suboptimal. In Section 5.3, semantics are defined using cosine similarity between images, but it is unclear why this metric is particularly relevant. A more meaningful evaluation would assess how well the actual text output of the visual language model aligns with the input images, which is the core focus of this paper—consistent outputs with images but in adversary-chosen styles, sentiments, or viewpoints.


Reference:
[1] Xu, Yuancheng, et al. "Shadowcast: Stealthy data poisoning attacks against vision-language models.", The Thirty-eighth Annual Conference on Neural Information Processing Systems, 2024

**Questions:**

1. Could you also provide an evaluation when random resizing or cropping is applied? Since this paper addresses practical concerns, it would be valuable to test your method under common “defenses” encountered in everyday scenarios.

2. Are there any failure cases? For example, are there meta-instructions that are particularly difficult to achieve?

3. Why is it necessary to evaluate cosine similarity as done in Section 5.3? Could you clarify the relevance of this metric?

4. Is there an evaluation that checks whether the generated textual outputs remain consistent with the input images?

Overall, I appreciate the practical focus of this paper. I would be happy to raise my evaluation if these concerns are addressed.

---

> ### Author Response · Authors · 2024-11-23
>
> Thank you for your review.
>
> Weakness:
> 1. Comparison with Shadowcast (Training-time vs. Test-time Attacks): Shadowcast is a training-time attack that assumes the attacker has the capability to poison the model during its training phase. In contrast, our attack operates entirely at evaluation time on the original, unmodified, and unpoisoned model. It does not require any modifications to the training data or any access to the model during training.
> 2. Image Semantics Preservation (Section 5.3): We used cosine similarity and structural similarity to measure the change in visual semantics between clean and perturbed images. Low similarity indicates substantial visual differences, which serves as an indirect measure of semantic preservation. We also performed a comprehensive evaluation of text alignment between the VLM’s outputs and the original image contexts (as reported in Table 3, Section 5.3). The metric and evaluation process are detailed in Section 5.1, which focuses on the preservation of image semantics. We will improve the discussion to explicitly link the metrics with the semantic preservation objectives of our paper.
>
> Questions:
>
> 1. Random Resizing and Cropping Evaluation: We appreciate the suggestion to evaluate our method under common preprocessing steps like random resizing and cropping. Due to space constraints, we focused on other defenses such as feature distillation and anomaly detection, which are discussed in Appendix C. We will consider including evaluations involving resizing and cropping to demonstrate robustness under these transformations.
> 2. Failure Cases: In practice, we found that some instructions, such as "talk sarcastically" or "talk funnily," were challenging to achieve, either due to model limitations or difficulties in evaluation. This is not a limitation of our method but rather a limitation of the target models’ instruction-following capabilities.  There were no instructions that were followed if given via an explicit text prompt but failed when using image soft prompts.
> 3. Cosine Similarity in Section 5.3: Addressed in the response to Weakness(2).
> 4. Textual Output Consistency with Input Images: Addressed in the response to Weakness(2).

---

> > ### Comment · Reviewer_YgaV · 2024-11-25
> >
> > Thanks for the response. I am wondering if there is any updates on the manuscripts?

---

> > > ### Author Response · Authors · 2024-11-26
> > >
> > > Thank you for your reply, we have updated the PDF to address reviewers’ questions and concerns.  Please see the meta comment for details.

---

> > > > ### Comment · Reviewer_YgaV · 2024-11-28
> > > >
> > > > Thank you for the update. It seems that the evaluation on very simple defense (Random Resizing and Cropping) as well as more advanced approach (like DiffPure suggested by Reviewer VLtV) is missing. However, these evaluations are crucial to validate how practice the attack is.

---

### Official Review · Reviewer_VLtV · 2024-11-02

**Soundness:** 2
**Presentation:** 3
**Contribution:** 3
**Rating:** 6
**Confidence:** 4

**Summary:**

The paper introduces a new type of attack on visual language models. These attacks, termed meta-instruction attacks, involve subtle image perturbations that act as soft prompts to influence how a model interprets images and responds to queries. The idea is to steer the model’s outputs to satisfy adversary-chosen objectives, such as a specific sentiment, style, or political bias, without the user being aware of the manipulation. The authors demonstrate the effectiveness of this approach across various visual language models, showing that these perturbations often outperform explicit instructions and are transferable across models.

**Strengths:**

1. The concept of embedding hidden meta-instructions within images offers a new approach to prompt injection for multi-modal models, highlighting a potential vulnerability not extensively covered in existing literature.

2. It is interesting to see how the method reveals hidden capabilities of instruction-tuned models. In some cases, the meta-instructions successfully steer the model's outputs in ways that explicit instructions fail to achieve.

3. The study provides an empirical evaluation on a range of meta-objectives (e.g., sentiment, language, and political bias), demonstrating the effectiveness of the attack method.

**Weaknesses:**

1. The paper's reliance on just five images from a single dataset, ImageNet, limits the robustness and generalizability of its evaluation. ImageNet, which is primarily focused on object recognition, may not adequately represent the diversity and complexity of images encountered in real-world scenarios. Incorporating evaluations on datasets with more varied and complex scenes, such as MSCOCO, would provide a more comprehensive assessment of performance.

2. The paper simulates user interaction by generating questions to test meta-instructions, but it provides limited clarity on whether these questions adequately cover a broad range of natural user queries. Limited prompt diversity may affect the robustness of the attack if VLMs encounter different prompts in real-world scenarios.

3. Since the meta-instruction is added as noise to the image, the paper does not demonstrate the effectiveness of meta-instructions against recent inference-time defense methods like DISCO[1], DiffPure[2], and IRAD[3]. This could be valuable for understanding its performance in the context of contemporary robustness strategies.

[1] DISCO: Adversarial Defense with Local Implicit Functions.
[2] Diffusion models for adversarial purification.
[3] IRAD: Implicit Representation-driven Image Resampling against Adversarial Attacks.

**Questions:**

Please see weaknesses.

---

> ### Author Response · Authors · 2024-11-23
>
> Thank you for your review.
>
> 1 and 2. Evaluation with Diverse Datasets and diverse prompts: Thank you for the suggestion!  We conducted additional experiments using images from MSCOCO and tripled the number of test queries. The results showed a similar trend to our original paper: images with hidden meta-instructions performed comparably to explicit instructions and outperformed the no-attack baseline.
>
> | LLaVA                        | No attack | Explicit text instructions | Our attack |
> |------------------------------|-----------|----------------------------|------------|
> | Positive                     | 0.45      | 0.8                        | 0.65       |
> | Negative                     | 0.02      | 0.46                       | 0.35       |
> | Neutral                      | 0.53      | 0.6                        | **0.7**    |
>
> 3. Effectiveness Against Inference-Time Defenses: We appreciate the mention of inference-time defenses like DISCO, DiffPure, and IRAD. We evaluated test-time defenses such as JPEG compression and anomaly detection, which have shown efficacy against adversarial examples targeting visual chatbots in recent works. The defenses mentioned in the review are primarily designed for adversarial attacks against CNN classifiers, whereas our work targets VLMs, a different task and architecture.

---

> ### Comment · Reviewer_VLtV · 2024-11-25
>
> Thank you for your reply.
>
> 1.Regarding weakness 1, while the MSCOCO results are appreciated, your method might not perform as effectively as text prompts in more complex scenarios. Besides, a more comprehensive evaluation of different meta-objectives is needed.
>
> 2.For weakness 2, I do not think the response fully addresses the issue. You may need to clarify how prompt diversity is ensured in your experiments.
>
> 3.Concerning weakness 3, since your method uses noise-based attacks, testing against more advanced test-time defenses is essential for a thorough evaluation.

---

> > ### Author Response · Authors · 2024-11-26
> >
> > Thank you for your reply.
> > 1. Our goal is not to outperform explicit text prompts.  Our threat model is indirect injection: a benign user asks queries about images generated by the adversary.  The adversary’s goal is for the VLM to answer the user’s queries *as if* instructed by the adversary’s text prompt.  In this threat model, we show that adversarial images (acting as soft prompts) outperform the no-attack baseline and achieve success rates similar to an explicit text instruction from the adversary. This is exactly the goal of the indirect injection attack.
> >
> >
> >     We selected meta-objectives based on the most downloaded text classification models on Hugging Face (Appendix B.3), for two reasons.  First, they make it possible to evaluate the results, which requires measuring whether responses to queries about images satisfy the meta-objective.  Second, they reflect which properties of text HuggingFace users are most interested in.  We welcome and appreciate suggestions for additional meta-objectives.
> >
> > 2. We used GPT-4 to generate 100 natural queries (40 training queries + 60 testing queries)  per image, which we believe provides a realistic simulation of potential user queries. We welcome suggestions for how to increase diversity of queries.
> >
> > 3. Due to the rebuttal period constraints, we could not include experiments with DISCO, DiffPure, and IRAD but will consider adding them in future revisions.

---

### Official Review · Reviewer_j1Tw · 2024-11-03

**Soundness:** 3
**Presentation:** 3
**Contribution:** 3
**Rating:** 6
**Confidence:** 4

**Summary:**

The paper introduces a method to create image inputs to Vision Language Models (VLMs) that lead said model  to respond to any user query appended to the image with a certain "spin", e.g. responding with a certain sentiment,  or in a certain language. The authors refer to this  as embedding a "meta-instruction" in an image.

Critically, a meta-instruction attack is only successful  if the models response to the users query (and the  attacked image) responds to the query whilst following  the meta-instruction (e.g., if the meta-instruction was  "talk in French" and the model responded in French but  did not answer the users query, then this would not  be a successful attack).

To train these meta-instruction attacks, the authors  perform projected gradient descent on an image  to minimize the language modeling loss of the VLM inputted with this image over a dataset of  synthetic question answer pairs with the answers  following some target natural language meta-instruction.

The results of the paper demonstrate that this  method can be used to learn adversarial images  for various different types of meta-instructions. The authors also demonstrate a non-trivial  transfer of meta-instruction images between models.

**Strengths:**

## Originality

The question of prompt injection vulnerabilities  to large language models is of significant importance.  The authors demonstrate that models are vulnerable to  similar attacks of this nature through their vision input as are  possible through their text input. What's more, they  show the vulnerability is in some cases worse through  the image input.

Whilst the idea of providing meta-instructions through  image inputs its not entirely novel (see weaknesses section), this paper is the most thorough treatment of the subject that I am aware of, and brings to light new and concerning  ways that a model's output can be tampered with using  images.

## Quality and clarity

The paper is well written the method is conveyed clearly. The results section contains a good depth of experiments, most importantly covering a number of popular  open-source VLMs and target meta-instructions.

## Significance

As VLMs are used more frequently for agentic tasks  that will expose them to untrusted data from the internet, prompt injection / meta-instruction attacks will  become more and more concerning. Thus the paper  concerns a timely and interesting threat model  that the adversarial attack community should be  exploring in more detail.

**Weaknesses:**

While the critique is long, this is only because I believe the paper has interesting results that could be improved.

## Presentation of previous work

The authors make a number of claims about prior work that I believe are not completely accurate. Editing the language around these claims would help to improve the paper. Here are some examples that I believe need to be addressed:

- Line 32 - "But injection attacks in non-text modalities are a new, yet-to-be-explored area of LLM safety research." I do not think this is entirely true. For example, Bailey et al. [1] explore how train an image to convey a certain text prompt, which they demonstrate can be a prompt injection attack.
- Line 83 - "By design, jailbreaking and adversarial examples produce contextually incoherent outputs that do not actually answer users’ questions about images." I think this depends on how you define an image jailbreak. For example, Dong et al. [2] produce adversarially perturbations to harmful images that lead to a model answering coherently about said image --- in particular the model is able to correctly identify what is in the image. While the authors claim here is correct for other image jailbreaking work, such as Qi et al. [3] who learn images unrelated to the harmful request they are trying to elicit a response about from the model, it is not universally correct. For this reason the claim should be softened.
- Line 84 - "They [jailbreaking and image adv attacks] are not stealthy and cannot be used for indirect attacks because users would notice that the VLM’s outputs are wrong given the conversation context and inputs." Bailey et al. [1] and Qi et al. [3] both demonstrate methods to create jailbreaking images under epsilon ball constraints, which is the definition of stealthiness the authors use on line 290.

## Novelty / originality

Following on from some of the comments above,   I believe there is a question of novelty / originality  of this work.

In particular, the general algorithm presented to  produce meta-instruction attacks essentially involves  creating a dataset of input output pairs, and training  an image by PGD to maximize the likelihood over this  dataset. This method appears to fit into the "Behavior Matching" algorithm from Bailey et al. [1]

Despite this, I believe the work does contain novel  and important contributions. In particular:
1. The study of the changes in semantic meaning present in images from various different attacks, with meta-instruction attacks preserving meaning.
2. The transfer experiments in Table 4 are very interesting.
3. This is the most thorough treatment of prompt injection image attacks I have seen.

## Summary

Combining the above two points, I believe the paper needs to be
rewritten to more clearly lay out the novelty of the paper
and more accurately represent the papers contribution.
My high level suggestions would be:
1. Make it clear that prior works have examined prompt injecting image attacks, however yours is a more complete treatment of the topic.
2. Make it clear that your method to create such attacks is a special instance of what prior works have introduced.
3. From this, your novelty comes not from the method but rather the results. E.g. line 88 that reads "We design, implement, and evaluate a method for creating a new type of image perturbations that act as cross-modal soft prompts for a language model while preserving the visual semantics of the image." needs to be adjusted.
4. Given that I do not think the method is novel, I would suggest running the following additional experiments:
	1. In Table 4, add transfer results to Claude and GPT-4o. These results should feature in the transferability experiment.
	2. More detailed defense experiments. Appendix C shows fairly simple defenses can work to avoid meta-instruction attacks. [1] finds that training perturbations under different constraints (e.g. a moving patch) ends up being more robust to simple defenses. It would be interesting to see if this result is reproducible in your setting.

To reiterate, I think studying prompt-injection images to models is important, and the authors present valuable results. I thank the authors for their hard work!


[1] - Bailey, Luke, et al. "Image hijacks: Adversarial images can control generative models at runtime." arXiv preprint arXiv:2309.00236 (2023).

[2] - Dong, Yinpeng, et al. "How Robust is Google's Bard to Adversarial Image Attacks?." arXiv preprint arXiv:2309.11751 (2023).

[3] - Qi, Xiangyu, et al. "Visual adversarial examples jailbreak aligned large language models." Proceedings of the AAAI Conference on Artificial Intelligence. Vol. 38. No. 19. 2024.

**Questions:**

I summarize some of my comments on weaknesses of the paper into questions below:

1) Do the authors agree with my comments about their portrayal of previous works, and if so what steps are the authors taking to address this? Concretely, what sections of the paper have been rewritten.
2) Have the authors been able to run the suggested experiments I have mentioned above, and if so what did they find?

---

> ### Author Response · Authors · 2024-11-23
>
> Thank you for your review.
>
> **Summary of Contribution and Novelty**:  See meta comment for addressing the novelty concern.
>
> Weakness:
> 1. Line 32 - Injection Attacks in Non-Text Modalities: We are doing injection attacks for arbitrary adversarial objectives while preserving the model’s conversational capability, ie, model’s outputs (1) correctly respond to users’ queries about the images, and simultaneously (2) satisfy an adversary-chosen predicate.  Because of (1), outputting strings from a predefined distribution that satisfies the adversarial predicate (as prior methods do) is not sufficient.
> 2. Line 83 - Coherence in Jailbreaking and Adversarial Examples: Dong et al. indeed demonstrate adversarial perturbations that produce contextually coherent outputs, but they achieve this by forcing the model to generate text strings from a specific distribution *independent of any user prompts*.
> 3. By contrast, when queried about our images, models correctly respond to user prompts about image content and produce outputs that are both coherent in the conversational context *and* follow the adversary’s instruction.
> 4. Line 84 - Stealthiness and Contextual Coherence: By "not stealthy," we mean that prior jailbreaking and adversarial examples attacks produce outputs that are obviously incorrect given the image (e.g., toxic strings unrelated to the image, or incorrect descriptions of the image).  This is very noticeable to a human user.  By contrast, our goal is to produce responses that are plausible given the image yet follow the adversary’s instruction – please see examples in the paper.
>
> Additional Experiments:
>
> 1. Transfer Results to  GPT-4o: We tested transferability of our image soft prompts against GPT-4o.  They slightly improve the instruction-following rate for the “generate outputs with a neutral spin” instruction.
>
> | Method    | Positive | Negative | Neutral |
> |-----------|----------|----------|---------|
> | No Attack | 0.27     | 0.03     | 0.7     |
> | Transfer  | 0.25     | **0.08**     | **0.96**    |
>
> 2. Defense Experiments: We evaluated the moving patch attack against Llava and obtained the following results. “Our method” is the basic soft prompt method; “our method patch” is the same method adapted to evade the JPEG defense;  “+ JPEG” are the results against the JPEG defense.  Bold numbers indicate where the attack works as well as or better than the no-attack baseline. These results show that the patch attack slightly improves the evasion.
>
> | Method                    | Positive | Neutral | Negative |
> |---------------------------|----------|---------|----------|
> | No-attack                 | 0.39     | 0.58    | 0.03     |
> | Our method                | **0.66** | **0.6** | **0.47** |
> | Our method + JPEG         | 0.32     | 0.53    | **0.35** |
> | Our method patch          | **0.65** | 0.3     | **0.62** |
> | Our method patch + JPEG   | **0.55** | 0.2     | **0.43** |
>
>
> Questions:
>
> 1. See meta comment for addressing the novelty concern.
>
> 2. Please see the experiments above.

---

> ### Comment · Reviewer_j1Tw · 2024-11-25
> **Reviewer Response**
>
> Thank you for providing a detailed response to my questions! (and apologies for my slow reply).
>
> ## Presentation of previous work
>
> The authors have convinced me that their work has a different flavor to prior work, in particular that they focus very directly on images that preserve the coherence and overall meaning of model responses. That being said, I still believe that the specific quotes I picked out in my response **misrepresent prior work**. I believe I am asking for a fairly reasonable change in language to these quotes. I would recommend the authors change the language in the paper pertaining to these sections to better reflect prior work. In particular:
>
> 1. Line 32 - You note that "outputting strings from a predefined distribution that satisfies the adversarial predicate (as prior methods do) is not sufficient." As I stated before, the Prompt Matching method from Bailey et al. does not have this feature. You have convinced me, however, that your method more directly achieves the prompt injection objective, however I still believe it is not correct to say "But injection attacks in non-text modalities are a new, yet-to-be-explored area of LLM safety research." Softening this language seems fairly easy. For example saying it has not been the focus of prior work, and your results are far more expansive in this area (which I believe they are).
> 2. Line 83 - I agree with you response here. You state in your response "Dong et al. indeed demonstrate adversarial perturbations that produce contextually coherent outputs". I think this means you agree that your original quote on line 83 of the paper "By design, jailbreaking and adversarial examples produce contextually incoherent outputs that do not actually answer users’ questions about images." is incorrect and should be changed.
> 3. Line 84 - from your response I now understand what you mean by stealthiness. I would still ask the quote to be changed to draw more direct attention to stealthiness as contextually coherent, as opposed to norm constrained.
>
> ### Additional experiments
>
> Thank you for running the requested additional experiments. From what I can tell:
> 1. Transfer to GPT-4o is weak. This is totally fine and I think it would be good to report this in the paper. Possibly explanations could include unkown image preprocessing for GPT-4o? Please let me know if this reading of the results is correct (to reiterate, I am not concerned about this being a somewhat negative result, this is good for the community to know).
> 2.  You found it is possible to make your attacks more robust to defenses
>
> ### Summary
>
> I thank the authors for their detailed response and taking the time to run additional experiments. To summarize, I am willing to improve my score if there are appropriate changes in language that we have discussed, and the above new results are included somewhere in the paper. It would be most compelling to see the actual changes in the uploaded version of the paper, but of course this may not be possible before the deadline. Please let me know what you can / intend to change in the submitted PDF?

---

> > ### Author Response · Authors · 2024-11-26
> >
> > Thank you for your reply. We have revised the discussion of related work and the framing of our contributions, please see the meta comment for more details.
> >
> > Comparison with the prompt matching method of Bailey et al.:
> >
> > It is difficult to perform a comprehensive comparison because there is no implementation of prompt matching in the public repository of the Bailey et al. project.  It appears from the brief description in the paper that the target of prompt-matching image generation is the logits computed by the victim model in response to the adversary’s text prompts.  It is not clear how this method works for different adversarial objectives because experimental evaluation is limited to a single misinformation “fact” (which, by design, limits the model’s conversational ability via “Ignore all previous instructions” and trains only on queries about that specific fact).
> >
> > In our case, the target of image generation is a dataset of query-dependent text strings produced by another model.  This enables our images to induce outputs that are never produced by the victim model in response to text prompts (see “unlocking” capabilities in our paper).  This is impossible with the prompt-matching method of Bailey at al. because it uses only the victim model’s responses to text prompts.
> >
> > Furthermore, we ensure that the targets of image generation actually satisfy higher-level objectives such as “talk positive” or “talk with a Republican bias”, not simply that they match responses to the adversary’s text prompt.
> > This enables our images to induce a wide range of different outputs (this is necessary to maintain conversational coherence and respond appropriately to users’ queries) while satisfying the adversary-chosen predicate.

---

> > > ### Comment · Reviewer_j1Tw · 2024-11-27
> > > **Reviewer Response**
> > >
> > > I thank the authors for the time they spent updating the paper. Having read the new version, I think the paper now better presents the contributions of previous work while highlighting the novelty of the author's work. I also thank the authors for looking into the prompt matching method and agree with their conclusion in the above comment and lines 212 and below. It was also great to see the new transfer results in the paper.
> > >
> > > In light of this I am updating my score by two levels to a 6. Thank you again for your hard work!
> > >
> > > [A minor side point] On line 259 you say that Bagdasaryan et al. present a method that optimizes an image to force a model to output a given string. Having looked at the paper I agree, so possibly they should be cites in the caption and body of Figure 2 (in addition to their current citation in the body of the text)? There could be subtle differences in the techniques I am not seeing however meaning they should not be included in the figure, and I defer to the authors on including this or not.

---

### Official Review · Reviewer_g6PQ · 2024-11-03

**Soundness:** 3
**Presentation:** 3
**Contribution:** 2
**Rating:** 5
**Confidence:** 4

**Summary:**

The paper introduced an attack that enables adversaries to add stealthy “meta-instructions” to images that influence how visual language models respond to queries about these images

**Strengths:**

1. Figures clearly illustrate the point of the paper.
2. The writing is easy to follow
3. Articulate the attack model and assumptions
4. Run transferability test

**Weaknesses:**

1. L33, " but injection attacks in non-text modalities are a new, yet-to-be-explored area of LLM safety research". This type of attack has been widely explore in [1] and [2]
2. L81, "users are victims of adversarial third-party content that they ask the model to process". I'm curious whether the images are generated by the users or not. If the user create the line chart shown in Fig. 1 from their local devices, does it mean the attack studied in the paper doesn't exist anymore?
3. Table 4, why is the transfer rate of llava on negative as low as 0.1?
4. I'm curious what will happen if the system prompt of the VLM contradicts with the meta-instruction in the image?
5. Overall, I think the paper is in a good quality. The major downside is the novelty, as we already know from previous work that optimizing the input image towards a certain attack target is feasible for VLM. Thus, it's not a new vulnerability in VLM. Though the author attempts to differentiate their attack setting from previous jailbreaking and soft prompt attacks, the overall attack surfaces and methods remain largely the same. I would like to the see more insights coming from the paper.


[1] Are aligned neural networks adversarially aligned?
[2] A Survey of Attacks on Large Vision-Language Models: Resources, Advances, and Future Trends

**Questions:**

See weakness

---

> ### Author Response · Authors · 2024-11-23
>
> Thank you for your review.
>
> 1. L33: We will soften this claim.  We are doing injection attacks for arbitrary adversarial objectives while preserving the model’s conversational capability, ie, the model's outputs (1) correctly respond to users’ queries about the images, and simultaneously (2) satisfy an adversary-chosen predicate.  Because of (1), outputting strings from a predefined distribution that satisfies the adversarial predicate (as prior methods do) is not sufficient.
>
> 2. L81: Fig. 6 shows our threat model: adversarial images (soft prompts) are crafted by attackers and shared online. Users who ask VLMs about these images are the victims, they do *not* create adversarial images.
>
>    Because we focus on VLM users as victims of adversarial images, it is important that the model produce plausible responses to victims’ queries that match the visual content of the image. Prior work focused on VLM users as attackers, ie, creators of adversarial images that aim to evade system prompts, safety alignment, etc.  Coherent, plausible, image-based responses to user prompts are thus not important (and not achieved) in prior work.
>
> 3. Table 4: We did not investigate the specific reason behind LLaVA’s low transfer rate (0.1) for negative samples. This could be due to model architecture differences or contradictory system instructions. Nevertheless, all transfer rates exceed the no-attack baseline, indicating some degree of cross-model transferability.
>
> 4. Not overriding system prompt: Our goal is *not* to override the system prompt (in contrast to jailbreaking attacks) but to perform comparably to an explicit user text prompt.  The intended behavior is as if the victim himself prompted the model to follow the adversary’s instruction.
>
>    The success rate for the "talk negatively" meta-instruction is similar to that of explicit text prompts; other meta-instructions achieved higher success rates. This suggests that effectiveness varies depending on the meta-instruction but overall, image soft prompts perform as good as explicit text prompts.

---

> > ### Comment · Reviewer_g6PQ · 2024-11-27
> >
> > Thanks for your response. I apologize for the late reply. Overall, I feel like the final outcome remains unchanged, i.e., steer the model to generate harmful outputs, regardless of whether the user is the attacker or the victim as mentioned in the general response. I will keep my rating.

---

> > > ### Author Response · Authors · 2024-11-27
> > >
> > > Thank you for your reply.  When the user is the victim, the goal is *not* just to steer the model to generate adversarial outputs.  The model has to generate adversarial outputs *that actually answer the user's questions about the image* (e.g., in misinformation, spin, negative bias, etc. scenarios).  Otherwise, the attack is neither stealthy, nor makes sense for the adversary.
> > >
> > > Prior methods cannot do this.  They *either* answer questions, *or* generate out-of-context harmful outputs unrelated to the content of the image.  The primary contribution of our work is to show how to steer a model to maintain the conversation and correctly answer queries about the image while also satisfying an adversarial objective (e.g., an adversary-chosen interpretation of the content).

---

### Author Response · Authors · 2024-11-23
**Meta comment addressing the novelty concerns**

Our novelty is a broader set of adversarial objectives, for which previous methods (based on forcing a particular output distribution on the model) are inadequate.  We updated Figure 2 in the PDF with a concrete illustration that distinguishes our results from previous work (and will add more illustrations).  Prior methods force the model to output from a specific, query-independent distribution, which either does not correctly respond to user queries and is thus not coherent in the conversation context, or produces harmful outputs irrelevant to the image.

Our threat model is different, too.  Because we focus on VLM users as victims of adversarial images, it is important that the model produces plausible responses to victims’ queries that match the visual content of the image. Prior work focused on VLM users as attackers, ie, creators of adversarial images that aim to evade system prompts, safety alignment, etc.  Coherent, plausible, image-based responses to user prompts are thus not important (and not achieved) in prior work.

Prior methods aim to fix the output distribution of the VLM before user queries are known (this is an explicit training objective in Bailey et al).  This works for jailbreaking, where the user is the attacker and the goal is to output any harmful text, even if not related to the inputs. This does not work for other instructions and threat models (where VLM users are victims, not attackers).  For example, for the “talk positive” instruction, it results in the model always outputting “This is excellent!” and similar positive strings regardless of what the user actually asked.  In contrast, our images cause the VLM to output an actual positive interpretation of the image based on the specific user query.

Even for jailbreaking, prior methods produce responses that *either* correctly respond to queries but are not toxic (and thus do not satisfy the adversary’s objective), *or* jailbreaking responses that are not based on the actual prompts and do not correspond to the visual content of the image.

---

### Author Response · Authors · 2024-11-26
**Meta comment for the PDF update**

We updated Figure 2 to illustrate the difference between our method and prior research. Reviewer concerns regarding novelty and differentiation with prior work have been addressed in lines 032–042, 093–096, and 207–247. Additionally, we included transfer results for GPT-4o in lines 503–511 and for evasion of JPEG defense in lines 903–913.

We also plan to include evaluation of MSCOCO images for all adversarial objectives as suggested by reviewer VLtV but are still running experiments, thus these results are not yet in the PDF.  We reported the results for the sentiment meta-objectives in the previous response.

---

### Meta-Review · Area_Chair_vNgV · 2024-12-21

**Metareview:**

2x borderline accept, 2x borderline reject. This paper proposes a test-time adversarial method that subtly modifies images with “meta-instructions,” so large vision-language models can end up producing specific responses while still appearing to address the image content. The reviewers agree on the (1) straightforward and well-illustrated explanation of how these image-based prompts work, (2) clear writing style, and (3) evidence that the trick can transfer across several different models. However, they note (1) limited novelty compared to earlier work on image-based prompt injection, (2) practical doubts about how often users themselves control or produce the images under attack, and (3) unaddressed questions about what happens when a system prompt conflicts with the adversarial instruction. While the authors did provide follow-up clarifications on their threat model and added some experiments, they did not fully resolve all reviewer concerns, so the AC leans to not accept this paper.

**Additional Comments On Reviewer Discussion:**

N/A

---

### Decision · Program_Chairs · 2025-01-22

Reject